

# The relationship between individual phenotype and the division of labour in naked mole-rats: it's complicated

James D. Gilbert, Stephen J. Rossiter and Chris G. Faulkes

School of Biological and Chemical Sciences, Queen Mary University of London, London, United Kingdom

## ABSTRACT

**Background**. The naked mole-rat (*Heterocephalus glaber)* is among the most social mammals on the planet, living in eusocial groups of up to 300 individuals that contain a single reproductive female and up to three reproductive males. A critical aspect of their complex social system is the division of labour that allows non-breeders to form an effective workforce. Age- or weight-based polyethisms are widely cited as explanations for how labour is divided, but evidence in support of these hypotheses has been equivocal.

**Methods**. To assess the extent to which individual working behaviour is determined by sex, age, weight and social rank, we studied the behaviours of 103 animals from eight captive colonies. We performed focal sampling and ran mixed-effects models to assess which factors explained variation in working behaviour during six ten-minute observation periods per individual.

**Results**. Contrary to widely-held beliefs, we found that working behaviour did not decrease linearly with weight, although polynomial regressions indicated younger and medium-sized individuals worked most frequently, while high-ranking individuals worked for the shortest periods of time. Working behaviour and its relationship with individual characteristics also varied between colonies.

**Conclusions**. While age- or size-based polyethisms may have some influence on working behaviour, we argue that other characteristics of the individual and colony are also important. In particular, the interactions of individual, social and environmental factors must be considered in order to understand the emergence and effectiveness of the division of labour that is so critical to many social organisms.

Corresponding authors
James D. Gilbert, j.gilbert@qmul.ac.uk
Chris G. Faulkes,
c.g.faulkes@qmul.ac.uk

## INTRODUCTION

The evolution of cooperation is a major driver of sociality and can provide the opportunity for labour to be divided so that individual contributions vary by task or by total effort (*Maynard Smith & Szathmáry, 1999*). The benefits of an effective division of labour have been studied in depth in social insects, in which a combination of behavioural flexibility and physical adaptations meet colony needs (e.g., *Wilson, 1987*; *Gordon, 1989*; *Jeanson & Weidenmüller, 2014*). In contrast, where tasks are divided in social mammal groups, the underlying causes of this division are poorly understood.
Naked mole-rats are widely considered to be eusocial mammals, characterised by an extreme reproductive skew in which 99% of individuals may never reach reproductive maturity (*Jarvis et al., 1994*). Breeders specialise in reproduction and, apart from offspring care, contribute little or nothing to general colony work (*Jarvis, 1981*). Non-breeders are responsible for the critical functions of foraging and nest maintenance, and evidence from the Damaraland mole-rat suggests the presence of additional non-breeders can increase the reproductive output of the colony (*Houslay et al., 2020*). In regions with low or unpredictable rainfall, the geophytes mole-rats feed on can be distributed unevenly, while the cost of digging through hard soil is high. Across the African mole-rat family, sociality is thought to mitigate the energetic cost and risks associated with foraging and dispersing in this habitat (*Faulkes & Bennett, 2013*), although whether cooperative foraging was a cause or consequence of eusociality is unclear (*Burda et al., 2000*). How work is distributed among non-breeders to maximise their output has not been fully explained. Researchers have proposed a number of ways in which tasks could be distributed effectively, often referring to an age- or weight-based polyethism, whereby individuals carry out different types of work according to their age or weight (*Jarvis, 1981*; *Jarvis, 1991*; *Faulkes et al., 1991*; *Lacey & Sherman, 1991*).

When *Jarvis (1981)* first described eusociality in naked mole-rats, she noted that non-breeders were responsible for all the collective work within a colony but did not contribute evenly. Specifically, she observed weight-based polyethisms in which heavier individuals worked less often, seemingly mirroring observations of social insect societies. This observation has grown into the widely-held belief that small individuals exhibit the most working behaviour, typically encompassing foraging, tunnel maintenance and care of young, while large individuals appear less active, and mainly serve as colony defence.

Since this early work, subsequent support for a 'caste-like' system in naked mole-rats has been mixed. For example, *Faulkes et al. (1991)* examined relationships among weight, sex and various working behaviours, and found evidence for small workers and large soldiers. Similarly, *Lacey & Sherman (1991)* reported that smaller non-breeders do work more, but, due to "considerable scatter" (pp. 322) in behaviours, they concluded that no distinct worker castes are present. *Jarvis, O'Riain & McDaid (1991)* also reported mixed results with respect to the relationship between weight and working behaviour. Yet despite these equivocal findings, the concept that naked mole-rat show size-based division of behaviour has become widely adopted as a rule-of-thumb (e.g., *Clarke & Faulkes, 1997*; *Jarvis, 1981*; *Jarvis, O'Riain & McDaid, 1991*; *Lacey & Sherman, 1991*; *Judd & Sherman, 1996*; *Hathaway et al., 2016*; *Mongillo et al., 2014*; *Jarvis et al., 1994*; *Lacey & Sherman, 1997*). A similar issue exists in social insects, where early reports of polymorphic ant workers with specialised roles has become a widely-held belief that working behaviour is closely linked to body size in general (*Gordon, 2016*). In reality, in naked mole-rats, while there has been some support for the relationship between body size and aggression towards conspecifics (*O'Riain & Jarvis, 1997*; *Mooney et al., 2015*), empirical evidence for smaller non-breeders exhibiting more active working behaviour has come from a single study since 1991 (*Jacobs & Jarvis, 1996*).
**Table 1** Summary statistics of animals used in the study.

| Colony | Total number of individuals (at start of observation period) | Number of individuals observed | Number of females observed | Proportion female | Mean (min, max) age (months) | Mean (min, max) weight (grams) | Mean (SD) working behaviour recorded (seconds per observation) |
|---|---|---|---|---|---|---|---|
| 11A | 28 | 15 | 6 | 0.40 | 30.7 (9, 77) | 27.5 (16.7, 41.9) | 47 (110.5) |
| 11B | 24 | 16 | 7 | 0.43 | 46.6 (18, 86) | 33.2 (23.5, 56.9) | 125.5 (197.5) |
| 11C | 16 | 10 | 8 | 0.80 | 33.1 (8, 63) | 27.8 (17.1, 50.4) | 73.7 (153.9) |
| 17A | 23 | 16 | 10 | 0.63 | 7.9 (4, 18) | 23.2 (12.8, 38.5) | 128.1 (180.7) |
| 800 | 10 | 8 | 7 | 0.88 | 100.5 (36, 184) | 30.5 (18.8, 38.6) | 103.1 (173.1) |
| CF05A | 10 | 7 | 2 | 0.29 | 26.1 (22, 41) | 33.6 (29.9, 45.6) | 178.3 (180.9) |
| CF27 | 26 | 18 | 13 | 0.72 | 65 (19, 154) | 28.6 (18.6, 44.2) | 203.6 (192.3) |
| FK100 | 43 | 13 | 5 | 0.38 | 38.1 (15, 93) | 25.5 (17, 48.4) | 59.7 (145.4) |
| Total | 180 | 103 | 58 | 0.56 | 41.9 (4, 184) | 28.3 (12.8, 56.9) | 116.6 (177.5) |

After almost two decades without further research, a recent analysis has raised further questions regarding earlier findings. *Mooney et al. (2015)* showed that digging behaviour was not associated with body mass or age, although aggression towards foreign mole-rats was positively correlated with body fat. In light of these results, we performed a large-scale study of the determinants of working behaviour in naked mole-rats. By observing individuals across eight captive colonies, we examined the effects of weight and age on working behaviour. Additionally, we tested for the effect of rank, a characteristic that could influence how much an individual contributes to collective activity but which has not been thoroughly tested (but see *Clarke & Faulkes, 2001*).

## MATERIALS & METHODS

We selected individuals from eight colonies of captive-bred naked mole-rats at Queen Mary University of London and the study was carried out in accordance with institutional guidelines. They were all captive born descendants of animals that were originally captured in Kenya during the 1980s. Our youngest subjects were four months old during observations, as naked mole-rats do not exhibit the full behavioural repertoire until around three months (*Jarvis, 1981*; *Jarvis, 1991*). Details of the individuals and colonies used in the study are given in Table 1 (Results, below). Each colony was housed separately in a single room, in a network of acrylic tunnels and boxes. The temperature of the room was maintained between 26–32 °C. Ambient light and noise were constantly recorded and changed unpredictably as people moved around the room and surrounding area. The animals were fed *ad libitum* every day. The diet generally consisted of carrot, sweet potato and butternut squash, although occasionally other fruits and vegetables were provided.

We observed all individuals in colonies that contained fewer than 20 in total, and randomly selected 20 individuals from each of the remaining colonies. The observation sequences of colonies and individuals within colonies were also randomly generated. If a colony's breeding female was not selected randomly, she was added to the colony's observation list manually (colonies 11A, 11C and FK100) as a means of comparison in our
focal studies. Each individual was observed three times in the morning and three times in the afternoon to avoid confounding variation from daily behaviour patterns. Animals were weighed in each of the three weeks they were observed, and means were calculated and used in the analyses. Animals were identifiable through RFID microchips and were marked twice a week with black marker pens so we could distinguish between them during observations.

Females were classified as breeders if they had a perforated vagina during the observation period, they were seen mating, or became pregnant, and we classified all other females as non-breeders. External genitalia are generally monomorphic in non-breeding naked mole-rats (*Jarvis, 1991*; *Peroulakis, Goldman & Forger, 2002*). However, the dark-red vaginal membrane can become prominent in some individuals, which is thought to reflect a partial release from reproductive suppression and may be a sign of incipient reproductive activation (*Jarvis, 1991*). We recorded which females had prominent vaginal membranes and excluded them from the analyses. Males seen mating were recorded as breeders and the rest were classified as non-breeders (*Jarvis, 1991*). While we recorded ano-genital nuzzling when possible, we did not classify animals as breeders if they were observed in these interactions as some ano-genital nuzzling involves non-breeders (*Jarvis, 1991*). Historical records of breeding status based on this classification were also used.

On entering the animal room, we allowed the colony to settle down from any increased behavioural activity for a period of up to 10 min and we minimised movement and noise while observing to avoid disturbing the animals. Behaviours were recorded between 08.00 and 18.00. Individuals were observed sequentially in their burrow system for ten minutes each, with each individual observed six times for a total of 60 min per individual. All observations were recorded in BORIS (Behavioral Observation Research Interactive Software) version 7.9.7 (*Friard & Gamba, 2016*).

The broad behavioural categories we recorded were as described in the ethogram of naked mole-rat behaviour: transport of food and nest material, digging and offspring tending (*Lacey et al., 1991*). As it was not possible to distinguish between digging and transporting behaviours, these were classified together as working behaviour. Offspring were only present in one colony during observations and the few instances of tending offspring were not included in the analyses.

We tested for an effect of dominance rank on working behaviour and aggression by establishing the dominance hierarchy within each colony using passing behaviour, which is a reliable indicator of dominance hierarchies in naked mole-rats (*Clarke & Faulkes, 1997*). We recorded which individuals passed over the top of other individuals during face-to-face encounters in tunnels. Interactions not thought to indicate rank include tail-to-face encounters, passing in chambers or corners of tunnels, when one individual digs throughout the encounter, and when individuals do not pass directly over the top of one another.

We used the Elo rating system to calculate individual dominance ranks within colonies (R package *EloRating, Neumann & Kulik, 2020*). The Elo rating method has several advantages over matrix-based methods such as its ability to calculate ranks within small groups and account for the loss of individuals during observations. After calculating each individual

rank, we scaled each rank to account for variation in group size by dividing by the number of individuals observed in the respective colony. Lower rank values (those towards zero) represent more dominant animals. We calculated the steepness of each colony's dominance hierarchy using the *steepness* function from the *EloRating* package, which is based on David's Scores (*De Vries, Stevens & Vervaecke, 2006*).

Ten-minute observation periods that contained no working behaviour were assigned a value of zero. To investigate which factors predicted whether an individual showed any working behaviour, we conducted logistic mixed-effect models, using the R package *lme4* (*Bates et al., 2020*, p. 4). For individuals that showed working behaviour in a given session, we determined the factors that predicted variation in the duration of working behaviour using linear mixed-effect models, also in *lme4*. For a discussion of "two-part" models, see (*Duan et al., 1983*; *Min & Agresti, 2002*). We checked the residuals of the logistic models for uniformity, dispersion, zero-inflation and the presence of outliers using the *DHARMa* R package (*Hartig, 2020*). For the linear models, we confirmed the residuals were normally distributed and had similar variances.

In both our logistic and linear models, we first constructed null models in which we included individual and colony fitted as random effects, with individual nested within-colony. The response variable was working behaviour per observation session. For each null model, we then constructed a set of separate models, each containing one of the individual characteristics as a fixed effect: sex, age, weight and rank. Due to the correlations between age, weight and rank (older individuals tend to be bigger and of higher rank, *Schieffelin & Sherman, 1995*; *Clarke & Faulkes, 1997*; *O'Riain & Jarvis, 1998*); we did not create models that contained more than one fixed effect. As relationships between work and age, weight and rank could be non-linear, we also created polynomial linear regressions that included the individual characteristic-squared and -cubed. We compared the performance of the models as described below and report the best performing for each characteristic.

Akaike Information Criterion (AIC) values estimate how well a model approximates the unknown reality relative to other models; smaller AIC values indicate better models (*Burnham & Anderson, 2002*). Second-order AICs (AICcs) were generated using the *aictab* function from the *AICcmodavg* package (*Mazerolle & Linden, 2020*) for each model and compared to see whether the inclusion of an individual characteristic reduced the AICc. The relative likelihood of a model given the data is calculated by $exp(-(\frac{1}{2})\Delta AICc)$, where $\Delta AICc$ is the difference in AICc between two models (*Burnham, Anderson & Huyvaert, 2011*). The ratios of model likelihoods can be used to calculate an evidence ratio, which indicates the extent to which the data support one model over another (*Burnham, Anderson & Huyvaert, 2011*). We report the AICc values, $\Delta AICc$s and likelihoods relative to the null model for each alternative model. Along with AIC values as indicators of relative model performance, we report Nakagawa's $R^2$ for each model to estimate how much variation in our working behaviour data is explained by the independent variables (*Nakagawa & Schielzeth, 2013*; *Nakagawa, Johnson & Schielzeth, 2017*), calculated using the *performance* R package (*Lüdecke et al., 2019*).

All statistical analyses were carried out in R Studio Version 3.6.0 (*R Core Team, 2014*) and figures were made using the *ggplot2* package (*Wickham, 2016*).

**Table 2  The average duration of working behaviour and percentage of time observed working per 600-second observation period for females, males and both sexes combined.**

| Sex | Average time observed in working behaviour (SD) | Percentage of time observed working |
|---|---|---|
| Female | 124.8 (179.2) | 20.8 |
| Male | 106.1 (174.9) | 17.7 |
| Combined | 116.6 (177.5) | 19.4 |

## RESULTS

We focally sampled a total of 133 Individuals. After excluding breeders, females with red vaginal membranes and individuals with missing sex, age, weight or rank data, a total of 103 individuals were used in the analyses. Each individual was observed six times, giving a total of 618 observation periods. Working behaviour was observed in 274 (44%) observation periods and no working behaviour was observed in 344 periods (56%). The average duration of working behaviour observed per ten-minute observation was 116.6 seconds, accounting for 19.4% of the total observation time. This is comparable to the 23.8% of time spent working by humans during a 40-hour week. Working behaviour per individual is broken down by sex, age, weight and rank in Figs. 1, 2, Tables 1 and 2, below.

To assess which characteristics predicted whether working behaviour was observed or not, we compared each of our logistic regression models with a single fixed effect to the null model and assessed model fit. $\Delta$AICcs showed that sex, weight and rank did not improve model fit ($\Delta$AICcs, versus null model: Sex $= +1.48$, Weight $= +0.79$, Rank $= -0.06$; model outputs in Table 3, SI5). Effect sizes and standard errors support the inference that these characteristics do not explain variation in whether an individual was observed working. In contrast, adding age was associated with a reduction in AICc value of 7.24 and the likelihood of this model was approximately 37 times higher than that of the null model. The model coefficient suggests individuals were less likely to be observed working as age increased and this result was significant at alpha $= 0.05$. While including weight did not improve the performance of the model (SI5), including weight-cubed reduced the AICc by 3.89 in a model that had a likelihood 7 times higher than the null model. Nakagawa's $R^2$ estimates suggest all models had limited explanatory power (conditional $R^2$ between 0.202 and 0.230), including those that performed better according to AICc, and more variation was explained by the colony and individual random effects than by any of the fixed effects (Table 3).

To assess which characteristics predicted the duration of work observed, we compared each of our linear models with a single fixed effect to the null model and assessed model fit. Linear models excluded observation periods during which no working behaviour was recorded. $\Delta$AICc values suggest adding sex, age, weight or rank as predictors did not improve the null model, which included only colony as a random effect ($\Delta$AICcs, versus null model: Sex $= +2.04$, Age $= +2.00$, Weight $= +1.28$, Rank $= +0.56$; model outputs in Table 4, SI5). Effect sizes and standard errors support the inference that these characteristics do not predict variation in the amount of time individuals were observed

Gilbert et al. (2020), *PeerJ*, DOI 10.7717/peerj.9891

**Table 3  Results of logistic mixed-effects models predicting whether working behaviour was observed or not.** The format of the models was: Presence or absence of working behaviour $\sim$ Colony|Individual + Fixed Effect where Presence of working behaviour is whether working behaviour was observed during the ten-minute observation period, Colony and Individual are random effects with Individual nested within Colony, and Fixed Effect was omitted in the null model and one of sex, age (months), weight (grams) and rank (0-1, scaled) was included in the corresponding models. The scripts used to run the models are available in File S3. Relative likelihoods are calculated as $exp(-(\frac{1}{2})\Delta AICc)$. Nakagawa's $R^2$ estimates the variance explained by the fixed effects (marginal variance) and both fixed and random effects (conditional variance) (*Nakagawa & Schielzeth, 2013*).

| Variable | Model | | | | | | | | | |
|---|---|---|---|---|---|---|---|---|---|---|
| | Null | | Sex (Base = F) | | Age | | Weight | | Rank | |
| | Co-ef | S.E. | Co-ef | S.E. | Co-ef | S.E. | Co-ef | S.E. | Co-ef | S.E. |
| Intercept | −0.28 | −0.30 | −0.21 | 0.31 | 0.10 | 0.34 | −0.28 | 0.30 | 0.02 | 0.36 |
| Sex | | | −0.16 | 0.21 | | | | | | |
| Age | | | | | −0.009 | 0.003 | | | | |
| Weight | | | | | | | −2.62 | 2.67 | | |
| Weight$^2$ | | | | | | | −4.58 | 2.49 | | |
| Weight$^3$ | | | | | | | 5.73 | 2.38 | | |
| Rank | | | | | | | | | −0.50 | 0.34 |
| Model AICc | 790.66 | | 792.14 | | 783.42 | | 786.77 | | 790.60 | |
| AICc versus null model | - | | +1.48 | | −7.24 | | −3.89 | | −0.06 | |
| Relative likelihood versus null model | - | | 0.48 | | 37.34 | | 6.99 | | 1.03 | |
| Nakagawa's $R^2$: | | | | | | | | | | |
| Conditional | 0.202 | | 0.202 | | 0.230 | | 0.210 | | 0.202 | |
| Marginal | 0.000 | | 0.002 | | 0.030 | | 0.024 | | 0.005 | |

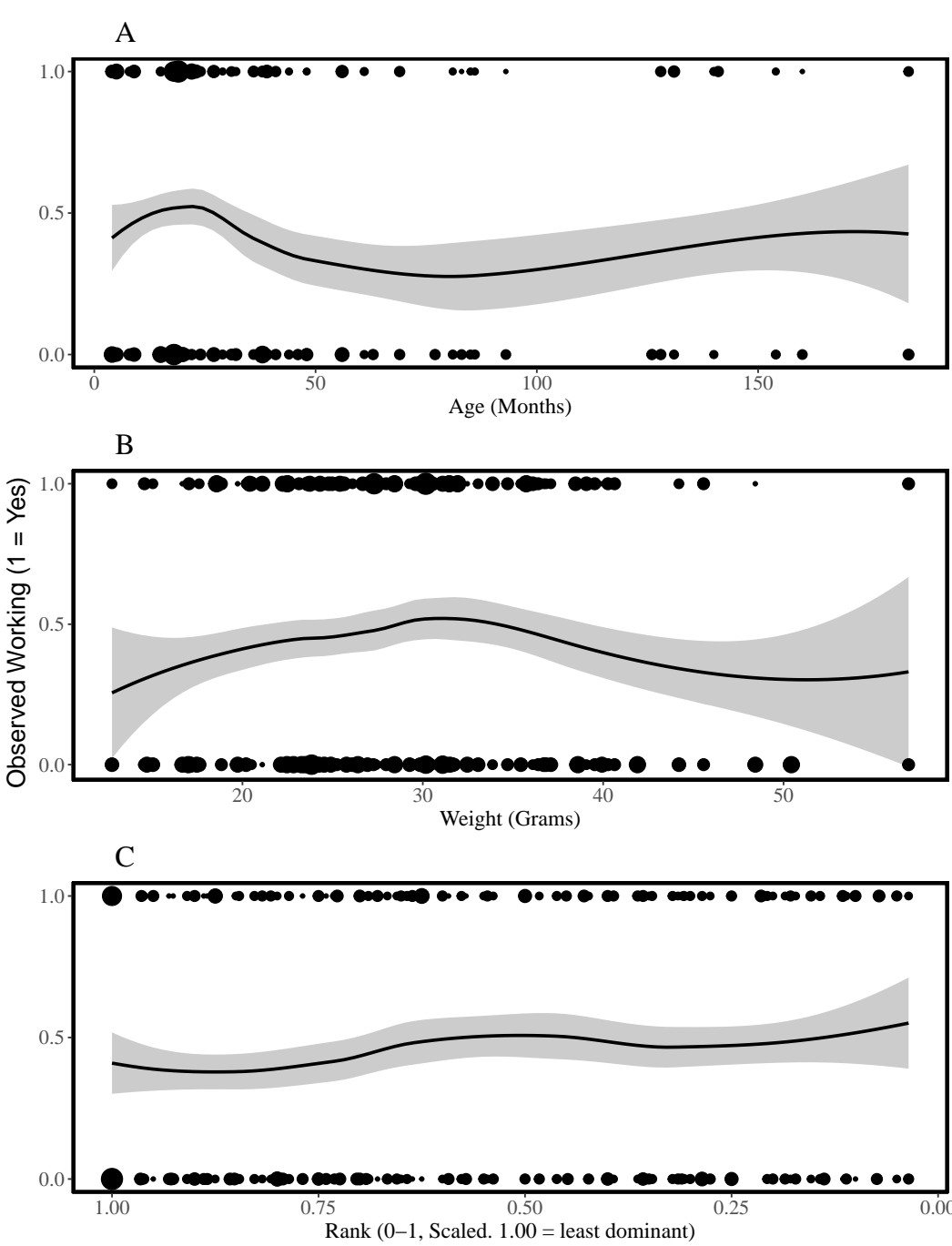

**Figure 1 Presence of working behaviour by age (A), weight (B) and rank (C) of the individual being observed.** The plots are fitted with locally weighted (loess) regression lines that display localised trends in the data. These are different from the models described below which include colony and individual as random effects. The size of the points reflects the number of data points at each location (geom_count, ggplot2). Rank is scaled to account for group size and the most dominant individuals have ranks closer to zero. Age and weight-cubed were associated with the probability that an individual was observed working; younger and mid-sized individuals worked most frequently.

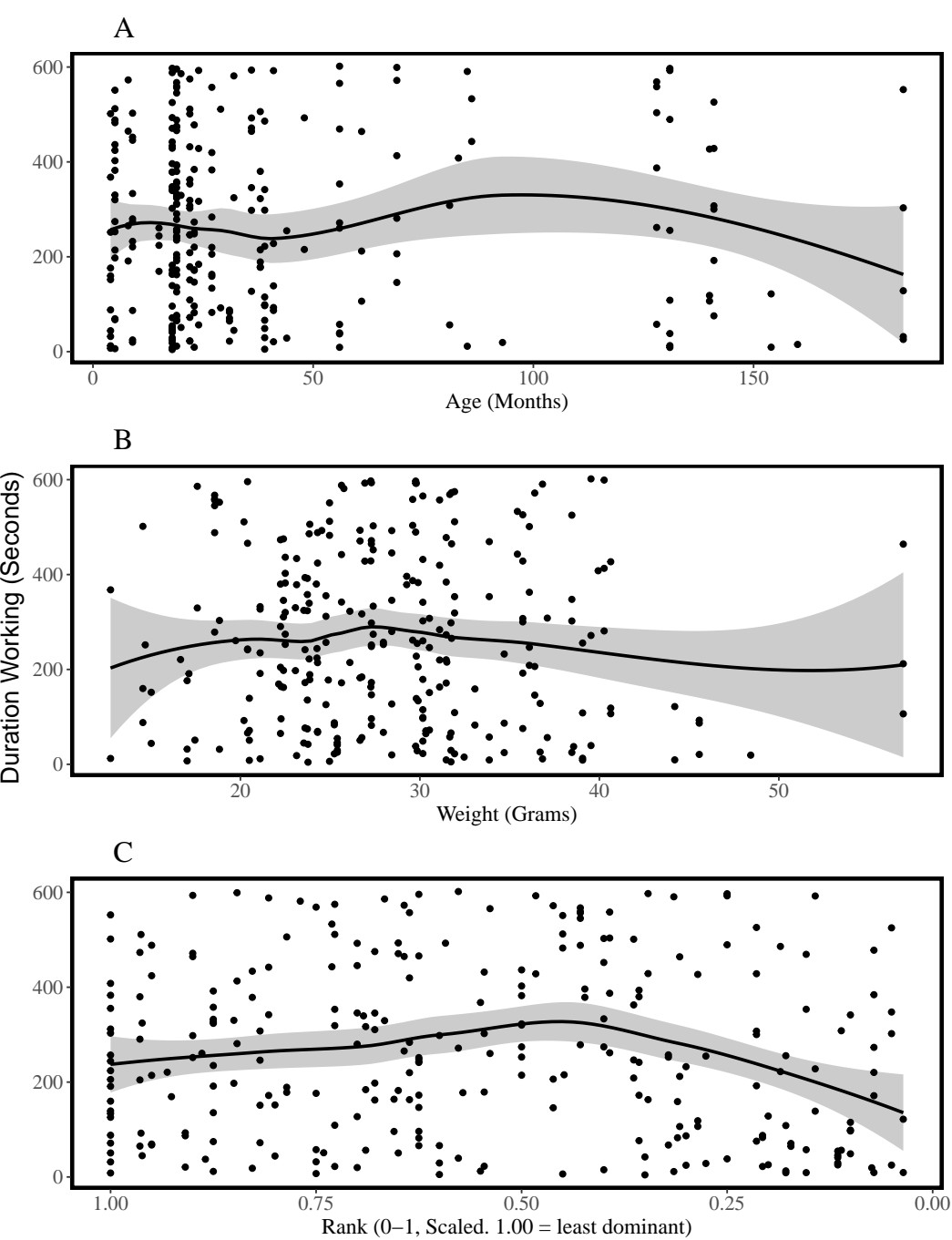

**Figure 2 Duration of working behaviour by age (A), weight (B) and rank (C) of the individual being observed.** The plots are fitted with locally weighted (loess) regression lines that display localised trends in the data. These are different from the models described below which include colony and individual as random effects. Rank is scaled to account for group size and the most dominant individuals have ranks closer to zero. Rank-squared was associated with the duration an individual was observed working; more dominant animals worked for shorter periods of time.

working. While including rank did not improve the performance of the model (SI5), including rank-squared reduced the AICc by 7.21 in a model that had a likelihood 37 times higher than the null model. Figure 2C suggests higher ranking individuals may work for shorter periods, although this effect appears to plateau and may even reverse among mid- and low-ranking animals. Nakagawa's $R^2$ estimates suggest all models had limited explanatory power (conditional $R^2$ between 0.169 and 0.182), including those that performed better according to AICc, and more variation was explained by the colony and individual random effects than by any of the fixed effects (Table 4).

In summary, age did predict whether an individual worked or not, but was not associated with the duration over which an individual worked. None of the other variables (sex, weight and rank) predicted either whether an individual was recorded working or not, or the duration of working behaviour. Weight-cubed also predicted whether an individual was observed working, while rank-squared predicted the duration of working behaviour. None of the models had high explanatory power and the majority of the variation in working behaviour was unexplained.

Plots of the duration of working behaviour and the relationships between working behaviour and individual characteristics within-colonies demonstrate the between-colony variation in our data (Figs. 3–5). Each colony's dominance hierarchy was assessed for how steep it was and assigned a value between 0 and 1, with 1 being the steepest. All steepness estimates were below 0.2.

## DISCUSSION

Previous studies indicate that some naked mole-rats spend more time working than others (*Jarvis, 1981*; *Lacey & Sherman, 1991*) and that differences between individuals may be stable for long periods of time (*Mooney et al., 2015*). Our analyses of 618 observation periods, encompassing 103 non-breeders from eight colonies, indicated that the probability of being observed working was explained by age and weight-cubed but not by sex, weight or rank. Initially, work frequency seems to increase with age and weight, until intermediate weights and around the age of two, after which the frequency of work plateaus and may even decrease (Fig. 1). The assignment of different roles to different age or weight classes distributes work across a colony without the need for costly cognitive evaluations of colony needs and worker availability based on patchy information (*Robinson, 1992*). Indeed, the presence of overlapping generations of offspring is considered a key criterion for eusociality (*Crespi & Yanega, 1995*) and different age and weight cohorts will almost always be present within eusocial groups. Despite the effect of age on probability of working, when we excluded periods in which no working behaviour was recorded, we found that only rank-squared predicted the amount of time spent working during the observation period. Figure 2C suggests low- and mid-ranking individuals worked for longer periods. Work duration was shorter among high-ranking, more dominant individuals.

In the absence of clear linear predictors, we suggest that age- or size-based polyethisms may depend on other conditions, such that colony members alter their working behaviour in response to the complex interaction of individual- and group-level pressures. Indeed,

Gilbert et al. (2020), *PeerJ*, DOI 10.7717/peerj.9891

**Table 4 Results of generalised linear mixed-effects models predicting the duration of observed working behaviour.** The format of the models was: Duration of working behaviour ∼ Colony|Individual + Fixed Effect Where Duration of working behaviour is the duration of working behaviour observed during the ten-minute observation period, Colony and Individual are random effects with Individual nested within Colony, and Fixed Effect was omitted in the null model and one of sex, age (months), weight (grams) and rank (0–1, scaled) was included in the corresponding models. The scripts used to run the models are available in File S3. Relative likelihoods are calculated with $exp(-(\frac{1}{2})\Delta AICc)$. Nakagawa's R2 estimates the variance explained by the fixed effects (marginal variance) and both fixed and random effects (conditional variance) (*Nakagawa & Schielzeth, 2013*).

| Variable | Model | | | | | | | | | |
|---|---|---|---|---|---|---|---|---|---|---|
| | **Null** | | **Sex (Base = F)** | | **Age** | | **Weight** | | **Rank** | |
| | **Co-ef** | **S.E.** | **Co-ef** | **S.E.** | **Co-ef** | **S.E.** | **Co-ef** | **S.E.** | **Co-ef** | **S.E.** |
| Intercept | 259.86 | 13.68 | 261.86 | 17.48 | 263.18 | 18.80 | 306.22 | 51.44 | 259.07 | 14.37 |
| Sex | | | −4.74 | 26.28 | | | | | | |
| Age | | | | | −0.09 | 0.31 | | | | |
| Weight | | | | | | | −1.66 | 1.73 | | |
| Rank | | | | | | | | | 282.05 | 200.03 |
| Rank$^2$ | | | | | | | | | −649.10 | 200.68 |
| Model AICc | 3622.44 | | 3624.48 | | 3624.44 | | 3623.71 | | 3615.23 | |
| AICc versus Null model | – | | +2.04 | | +2.00 | | +1.28 | | −7.21 | |
| Relative likelihood versus null model | | | 0.36 | | 0.37 | | 0.53 | | 36.78 | |
| Nakagawa's R$^2$: | | | | | | | | | | |
| Conditional | 0.169 | | 0.169 | | 0.171 | | 0.175 | | 0.182 | |
| Marginal | 0.000 | | 0.000 | | 0.000 | | 0.005 | | 0.056 | |

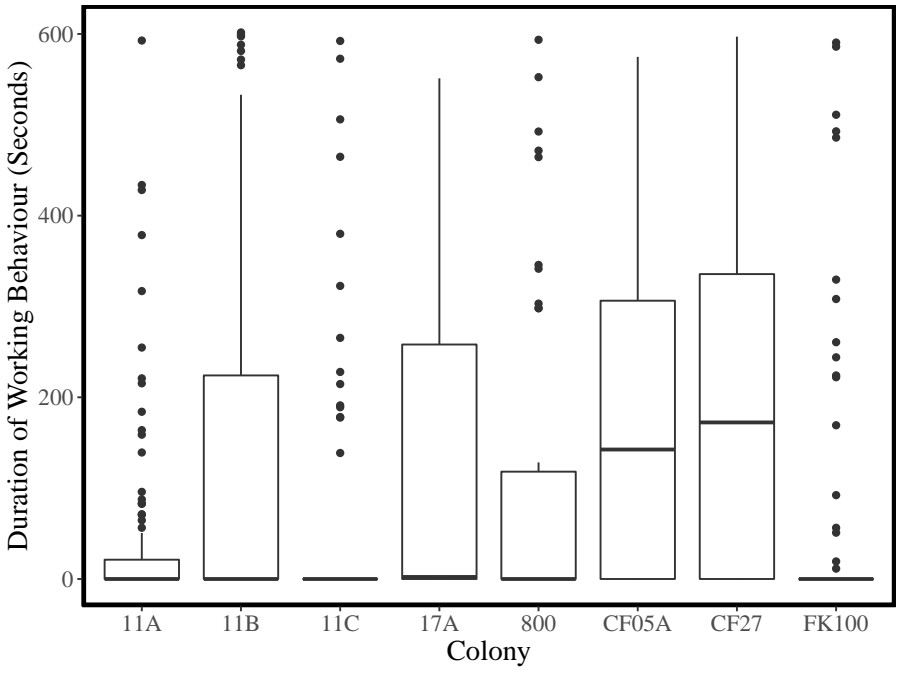

**Figure 3 Duration of working behaviour per observation period for each colony.** Medians are indicated by thick black horizontal lines, the upper and lower quartiles are represented by the upper and lower edges of the boxes, and the whiskers extend to 1.5 the interquartile ranges. Data points outside the whiskers are plotted as points.

*Mooney et al. (2015)* found that individuals changed their behaviours to compensate for the removal of their colony mates, indicating a level of flexibility in response to the needs of the group. Moreover, a relationship between age and behaviour was only evident when frequent-workers were removed, for which younger individuals compensated by increasing their work rate (*Mooney et al., 2015*). On the other hand, in the eusocial Damaraland mole-rat (*Fukomys damarensis*), age does seem to play a key role in cooperative behaviour (*Zöttl et al., 2016*; *Thorley et al., 2018*). Both studies of Damaraland mole-rats found helping behaviours increased until the age of one, after which point there was either a plateau (*Zöttl et al., 2016*) or reduction (*Thorley et al., 2018*) in helping behaviour. The similarity of helping behaviour in Damaraland and naked mole-rats is interesting, and implies convergent evolution of these patterns given that sociality is thought to have evolved separately in the two species (*Faulkes & Bennett, 2013*).

Taken together, our findings imply that older individuals perform fewer bouts of working behaviour, although these bouts do not differ in duration compared to those of younger individuals. In mammals, older individuals tend to be less active (*Ingram, 2000*; *Marck et al., 2017*) and our results may reflect this general mammalian trend. There is also evidence activity decreases after the of age three in the Ansell's mole-rat (*Schielke, Begall & Burda, 2012*). The relatively small number of working bouts seen in older mole-rats might also relate to them having other roles, which were not recorded here. For example, while

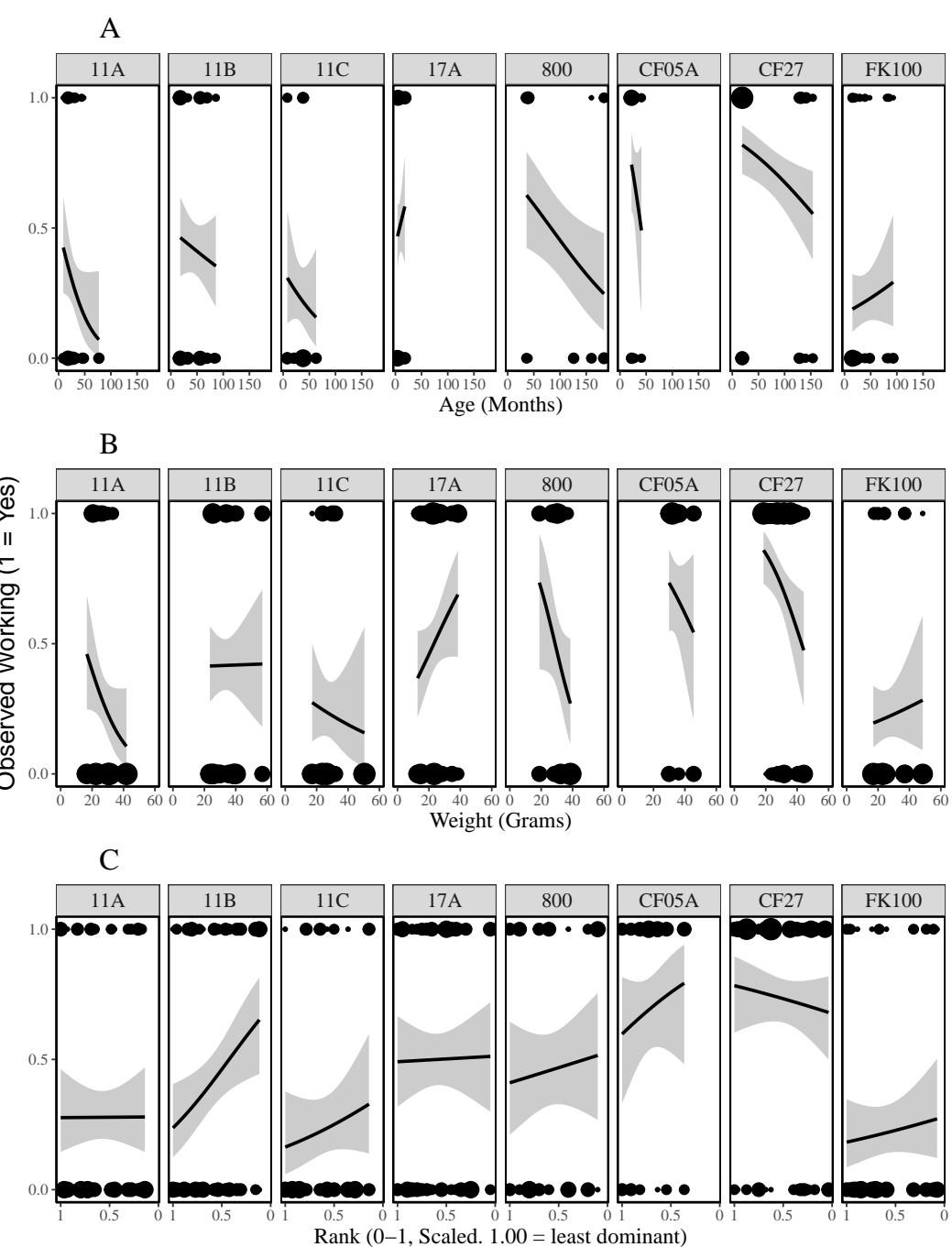

**Figure 4** **Presence of working behaviour by age (A), weight (B) and rank (C) of the individual being observed for each colony.** The lines are generalised linear regressions with a binary response variable (whether work was observed during the observation period) using ggplot2's stat_smooth function (*Wickham, 2016*). The size of the points reflects the number of data points at each location (geom_count, ggplot2).

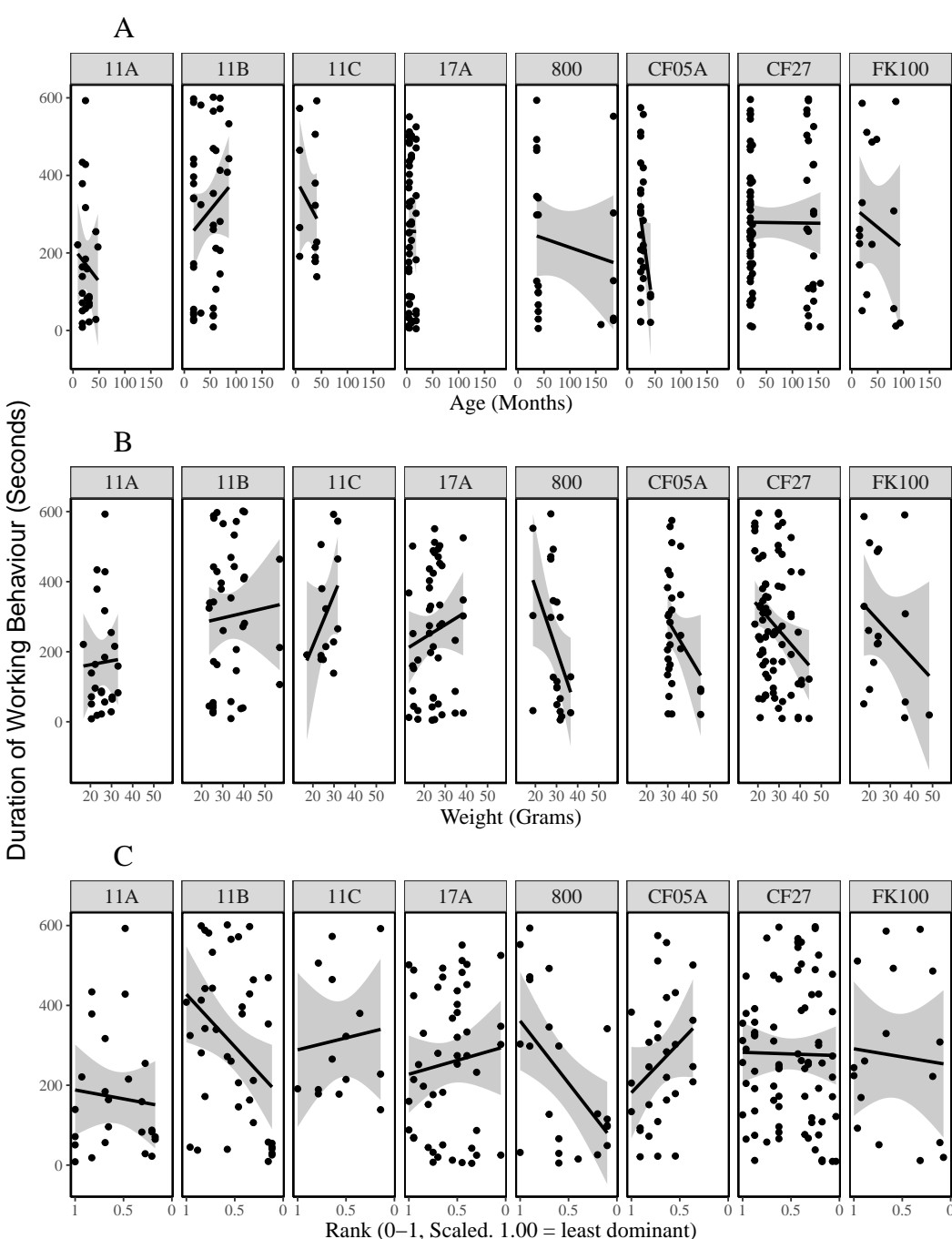

**Figure 5** **Duration of working behaviour by age (A), weight (B) and rank (C) of the individual being observed for each colony.** The lines are general linear regressions with duration of working behaviour as the response variable using ggplot2's stat_smooth function (*Wickham, 2016*).

we did not analyse aggressive or defensive behaviours, it is possible that older individuals, who also tend to be larger (*O'Riain & Jarvis, 1998*), spend more time defending the colony (*Mooney et al., 2015*). Older individuals may also invest less in working behaviour in order to maintain energy reserves for future challenges for dominance within the colony, particularly given that these animals are more likely to succeed in the event that a breeder dies or is removed (*Reeve, 1992*; *Clarke & Faulkes, 1997*; *Clarke & Faulkes, 1998*).

We found no evidence that work decreases linearly with body size, but work frequency increases with size until intermediate body weights are reached. After this, work frequency plateaus and may decrease. However, we are cautious in our interpretation of this result due to the relatively sparse data from animals of small and large body weight, and the tendency of polynomial models to overfit data (*Lever, Krzywinski & Altman, 2016*).

Across the African mole-rats, the link between age, size and working behaviour is still far from clear. For example, in naked mole-rats, *Mooney et al. (2015)* also reported no relationship between body size and working behaviour, although larger individuals were more aggressive towards conspecifics (*O'Riain & Jarvis, 1997*). On the other hand, body size does seem to be a good predictor of space-use in the Ansell's mole-rat (*Šklíba et al., 2016*). In contrast, a reported association between size and working behaviour in the social Micklem's mole-rat (*Fukomys micklemi*) was attributed to an underlying age-based polyethism (*Van Daele et al., 2019*), while a similar association in the Damaraland mole-rat may relate to growth rates (*Zöttl et al., 2016*). An early report on Damaraland mole-rats found that heavier individuals worked more (*Gaylard, Harrison & Bennett, 1998*). In this respect, it is interesting to note that naked mole-rats exhibit unusual growth patterns; growth rates vary between litters and individuals can gain size rapidly in response to social changes, even after extended periods of constant size (*O'Riain & Jarvis, 1998*). This decoupling of size and age may further complicate the relationship with behaviour.

We found no linear effect of social rank on working behaviour, but did find an effect of rank-squared. Naked mole-rat groups have strict dominance hierarchies (*Clarke & Faulkes, 1997*) and periods with high rates of aggression and fighting during which individuals are often killed (*Clarke & Faulkes, 1997*; *Clarke & Faulkes, 2001*; *Medger et al., 2019*). High-ranking naked mole-rats are more likely to become breeders if the previous breeders are removed (*Clarke & Faulkes, 1997*; *Clarke & Faulkes, 1998*), and may reduce energy spent on working behaviour in order to prepare for future dominance challenges. Alternatively, high-ranking individuals tend to be older and heavier (*Clarke & Faulkes, 1997*; *Clarke & Faulkes, 1998*) and may contribute less to general working behaviour but more to colony defence, as has been reported elsewhere (*O'Riain & Jarvis, 1997*; *Mooney et al., 2015*). More data are need before we can be certain about the distribution of work among heavier, older and higher-ranking individuals. In our study, we recorded an average of four interactions per individual, whereas the Elo-rating method for assigning rank performs better when there are around ten interactions per individual (*Sánchez-Tójar, Schroeder & Farine, 2018*). Thus, given the large confidence intervals around estimates of rank, we would like further evidence to confirm the impact of rank, which might be more evident over longer observational periods.

Our observations reveal considerable variation in the total amount of working behaviour and the relationships between work and individual characteristics within each colony (Figs. 3–5). With data from eight colonies, we did not have the power to test colony-level hypotheses, however, individual behaviours may be affected by a number of colony-level factors, such as food availability (*Reeve, 1992*), worker availability (*Mooney et al., 2015*) or the age-structure of the colony (*Gaylard, Harrison & Bennett, 1998*; *Šklíba et al., 2016*). Many published results from naked mole-rat behaviours have come from studies of one or a few colonies and were thus unable to account for colony-level variation. In this study, more variation in working behaviour was explained by individual and colony variables than by any individual characteristics. Our findings indicate that future research should increase the number of colonies used and statistically control for colony-level effects wherever possible.

This study looked at the factors that could predict whether an individual works and the variation in the amount of time individuals spend working. Labour could also be divided by task. Task specialisation has been observed in some social insect species (*Charbonneau & Dornhaus, 2015*) and, although it could be an important benefit to sociality in mammals, it has rarely been recorded (*Stander, 1992*; *Gazda et al., 2005*; *Hurtado, Fénéron & Gouat, 2013*; *Gazda, 2016*). *Mooney et al. (2015)* found that the amount of time spent on different tasks was stable in naked mole-rats, although *Thorley et al. (2018)* found no evidence for task specialisation in the eusocial Damaraland mole-rat. We could not test worker specialisation due to the lack of pup tending observed. Although some authors have shown that repeated exposure to a task increases efficiency in insects (*Langridge, Sendova-Franks & Franks, 2008*), this is not always the case (*Dornhaus, 2008*; *Santoro, Hartley & Lester, 2019*). Future research could use experimental tasks to explore the presence of specialisation in naked mole-rats and determine whether it increases individual or colony efficiency. Assessing worker, pup tending and defence behaviours like *Mooney et al. (2015)* would be the best way to establish whether task specialisation exists, as has been suggested in studies of other mole-rats (*Van Daele et al., 2019*).

## CONCLUSION

*Gordon (2016)* suggested that the term 'division of labour', implying internally-driven choices, is too rigid, and instead terms relating to individual behaviours should emphasise the influences of external conditions and social interactions. Further, we agree that workers should not be classified according to discrete 'castes' in mole-rats unless clear supporting evidence is reported (*Šklíba et al., 2016*). Polyethisms based on individual characteristics could be the start of an effective division of labour but the relationship appears to be complicated and variable. The outstanding questions are the extent to which polyethisms can be fine-tuned and the mechanisms that facilitate these changes. To date, the causes of within-individual variation are still largely unstudied in social mammals and insects (*Jeanson, 2019*). Given our results, combined with those of *Mooney et al. (2015)* and the mixed findings of earlier studies (*Faulkes et al., 1991*; *Jarvis, 1991*; *Lacey & Sherman, 1991*), future research should focus on the interaction between internal, social and environmental
influences on working behaviour, rather than attributing such strong influence to internal factors. As *Gordon (2016)* argues, interchangeability and the absence of fixed specialisation are exactly what make collective behaviour flexible and adaptive.

### Funding
This work was supported by the Natural Environmental Research Council (Grant NE/L002485/1). The funders had no role in study design, data collection and analysis, decision to publish, or preparation of the manuscript.

### Grant Disclosures
The following grant information was disclosed by the authors:
The Natural Environmental Research Council: NE/L002485/1.

### Competing Interests
The authors declare there are no competing interests.

### Author Contributions
- James D. Gilbert conceived and designed the experiments, performed the experiments, analyzed the data, prepared figures and/or tables, authored or reviewed drafts of the paper, and approved the final draft.
- Stephen J. Rossiter and Chris G. Faulkes conceived and designed the experiments, authored or reviewed drafts of the paper, and approved the final draft.

### Animal Ethics
The following information was supplied relating to ethical approvals (i.e., approving body and any reference numbers):

The research we did in this study involved observing, weighing and marking animals. These activities are all part of standard husbandry practice and are not regulated procedures covered by the 1986 Animals (Scientific Procedures) Act. All research was carried out in accordance with the institutional guidelines of Queen Mary, University of London.

### Data Availability
The raw data, code, and model output are available in the Supplemental Files.

### Supplemental Information
Supplemental information for this article can be found online at http://dx.doi.org/10.7717/peerj.9891#supplemental-information.

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
