# Peer review of "The relationship between individual phenotype and the division of labour in naked mole-rats: it’s complicated"

_PeerJ, doi:10.7717/peerj.9891_

## Round 0.1 · original submission · Minor Revisions

Thank you for your MS. Two reviewers and I enjoyed reading it. They have minor comments that you should address. Thanks!

Reviewer 1 ·

Basic reporting

This paper tests the hypothesis that in captive colonies of cooperatively breeding, eusocial naked mole rats, an individual’s traits, including age, determine whether or not it will work, and the duration of work undertaken. While the paper does not deliver a clearly significant outcome enlightening our understanding about which individuals work in such colonies, it provides evidence that increasing age rather than weight or behavioral rank determine how often a naked mole rat works, regardless of sex. The authors, however, realize the limitations of their captive study working with relatively small, fixed size colonies, and highlight that their data suggest a potentially malleable involvement in colony work depending on prevailing conditions and colony size. This paper will make a useful addition to the cooperatively breeding animal literature.

Experimental design

p.5, Material and Methods, 1st paragraph. Include here how many colonies studied comprised fewer than 20 members, and how many comprised more. It is not clear from Table 1.

p.5, 1st paragraph. “Where colony samples did not include a breeding female, one was selected (colonies 11A, 11C and FF)”. Please add the details for colony FF to Table 1. And, does this mean if the randomly selected individuals from a colony did not include the queen, she was added to that colony’s observation list anyway? It is not clear.

p.5, 2nd paragraph. “Red vagina” means closed, non-perforated external genitalia? Or can a non-perforate vagina be observed externally in naked mole rats?

p.5, 2nd paragraph. Provide citations previously validating the use of male mating or copulating (is this really just mounting, only?) and nuzzling female (queen?) genitals? If none, justify why these behaviors reliably identify male breeders.

p.5, 3rd paragraph. Maybe change “alarm behaviors” to “increased behavioral activity”? How were observers positioned relative to the colony under observation? In other words, could the colony detect vibrations from laptop key presses?

p.5, 4td paragraph, and p.6, 1st full paragraph. These basic methods should start the methods section at the top of p.5.

p.7, 2nd paragraph. Provide citation showing that naked mole rats with greater body weights are older.

p.8, Results. Replace “lucky enough to work” with “during”.

p.9. Fig. 1A-1C. The visual image of 0 or 1 values contributing to a best-fit line of some sort is not helpful to the reader. For a visual image, in addition to the modeling performed, could the authors substitute the ratio of working to non-working animals by age, weight and rank, e.g., if at 1 year of age 20 animals worked and 3 did not, that value could be represented by a ratio of 20/3 or 20/23? If no exercise like this is helpful, then the authors should only state values in the text or an extended Table 3.

Validity of the findings

p.3, Introduction. The authors need to also introduce some rationale for potential benefits to individuals to cooperate that may include kin selection, and provide some explanation as to why division of labor evolves during natural selection for cooperative breeding. Such added arguments would provide more justification for the study and its design.

Additional comments

All comments are entered above.

·

Basic reporting

The paper breaks down one established myth about the naked mole-rats and as such is worth publishing. There are, however, some methodical and interpretational problems and the literature is insufficiently covered. While the methodical problems can probably hardly be removed, they should be nevertheless discussed. The authors should also more critically revise (and reword or complement) some points of their interpretations and complement the references. While the authors break down one myth, they uncritically adopt others.

Terminology:
Which evidence is there that a female mates with up to three males, apart from a single study by Jarvis (1981) done on one incomplete family? Please provide references.
The authors introduce a new – in no way supported – category "possible breeders". This is rather absurd. Is there any evidence that whatever sexually mature naked mole-rat would not breed, i.e. would be an "impossible breeder", if given an opportunity?
Note that "colony", though widely used in literature, is not a correct term to designate a social group of naked mole-rats. A group like this, should be called in mammals a "family". A colony is a group of e.g. ground squirrels or seals which aggregate but are principally of solitary habits.
The authors are discussing "red vagina" as a trait characterizing breeding females. This is anatomically not correct. What they observe is "vulva" or, at best, vagina opening. The vagina is the tube between the vulva and the cervix.

Methods and interpretations:
Six times ten minutes observational time per one animal is a rather insufficient sampling. Particularly if sampled randomly, irrespective of any activity pattern. Both measures of activity do not correlate what the authors explain through their different characters (number of bouts versus length of bouts). I agree – note that it is a general characteristics of mammals (including human) that older individuals have fewer bouts of activity and these have longer duration than juveniles.
It would be much better and more informative to express both activity measures in per cent for every animal. I understand that N would not be so impressive (600+). The authors use per cent in their mix models, unfortunately not in figures.
The studied families differ greatly in their age structure. Two of them involve only young animals. It has been reported in the literature that young (fledgling) families behave differently. The authors should have focus on the largest, and with respect to age mostly differentiated families and observe all individuals in them.
The large figures at the end of the manuscript often show that the curve is pulled in a certain direction only thanks to a few outliers, otherwise there is no apparent trend.
The authors state (lines 230-233): Our analyses of 618 observation periods, encompassing 103 non-breeders from eight colonies, indicated that the probability of being observed working was explained by age - with older individuals working less - but not by sex, weight or rank. I do not fully agree that with this conclusion. Regarding age, this trend is apparent in six from eight families, regarding body weight in five out of eight families! Moreover inspite of diverse growth rate differences, age and body size always somehow correlate: youngsters are always smaller.

Work with literature:
It is strange that the authors cite and discuss studies on ants, wasps, lions, dolphins etc., but ignore papers on sociality in the next relatives of the naked mole-rats, particularly on Fukomys anselli and F. mechowii, published by the Czech and German teams which also address the age-polyethism, starting with the papers by Burda published in 1989 and 1990 and closing with the most recent of those studies: the paper by Skliba et al. (2016) in Scientific reports entitled: "... Ansell’s mole-rat indicates age-based rather than caste polyethism". This paper shows clearly that the main factor for any work (activity) ist the family-context – if there is less food and hard soil, all family members must work.
Moreover, a negative relation between body weight and digging work in mole-rats has been reported in the literature. However, in some families it may be not apparent, due to the context (motivation, requirements of the family). This was found i.a. by Lacey and Sherman (1991), as well as by two relevant papers ignored in the present manuscript: Gaylard et al. (1998), Skliba et al. (2015).
The authors define eusociality as used for arthropods by Crespi and Yanega (1995) (not Yanegra as written by authors) and ignore a highly cited paper by Burda et al. (2000) discussing eusociality in naked mole-rats.

In conclusion:
I believe the conclusions of the authors regarding the best explanation of differences in working frequency being the age-polyethism. Yet, the authors should more discuss also the effect of motivational context, as pointed out in papers by Lacey and Sherman (1991), Gaylard et al. (1998), Mooney et al. (2015), Skliba et al. (2016). The authors should include into the discussion more relevant papers by other authors on Fukomys mole-rats, rather than (or additional to) papers on insects, dolphins, lions etc. The authors should mention and critically discuss some shortcomings in the methods and interpretation. The paper can thus be improved by editorial work without necessity to do further experiments.

Experimental design

no further comment - please note that I reported about this aspect above in context of the whole contribution

Validity of the findings

no further comment - please note that I reported about this aspect above in context of the whole contribution

Additional comments

no further comment - please note that I reported about this aspect above in context of the whole contribution

---

## Round 0.2 · accepted · Accept

Thank you for your manuscript. After reviewing your revisions I believe you have fixed the issues highlighted by the reviewers. Thanks for a nice contribution.

PS: as a social insect biologist, I particularly enjoyed reading your article.

Reviewer 1 ·

Basic reporting

In this revised manuscript, the authors have clearly and thoughtfully responded to previous review concerns. I have no additional concerns.

Experimental design

Clear and rigorous, well described.

Validity of the findings

No comment.

Additional comments

The revised manuscript makes a more convincing case concerning the complexities of understanding which individual naked mole rats contribute to "working", for how long and in what ways. Thus will make a useful addition to the literature demonstrating even in a unique example of highly evolved eusociality in placental mammals, the mechanistic understanding of how eusociality is maintained is highly complex.